# Studies on the Ethyl Carbamate Content of Fermented Beverages and Foods: A Review

**DOI:** 10.3390/foods14193292

**Published:** 2025-09-23

**Authors:** Valentina Simion, Valerica Luminiţa Vişan, Ricuţa Vasilica Dobrinoiu, Silvana Mihaela Dănăilă-Guidea

**Affiliations:** 1Department of Biotechnologies, Faculty of Biotechnology, University of Agronomic Sciences and Veterinary Medicine of Bucharest, 59 Marasti Street, 011464 Bucharest, Romania; vali.simion@bth.usamv.ro (V.S.); ricuta.dobrinoiu@usamv.ro (R.V.D.); silvana.guidea@biotehnologii.usamv.ro (S.M.D.-G.); 2Industrial Biotechnologies Department, Faculty of Biotechnology, University of Agronomic Sciences and Veterinary Medicine of Bucharest, 59 Marasti Street, 011464 Bucharest, Romania

**Keywords:** ethyl carbamate, natural compounds, distilled alcoholic beverages, fermentation, healthy beverages, potentially carcinogenic

## Abstract

Ethyl carbamate, a genotoxic chemical contaminant present in fermented alcoholic beverages and foods, is formed from naturally occurring substances in these beverages and foods. Studies have shown that the content of ethyl carbamate can increase significantly during product storage and maturation, especially if favorable conditions are present. Higher levels of ethyl carbamate have been associated with distilled alcoholic beverages, mainly obtained from stone fruits. Ethyl carbamate content is lower in fermented foods, such as bread, yogurt, and fermented sauces. EC formation occurs through several different pathways in food systems. A primary pathway involves select compounds reacting with ethanol (EtOH); therefore, the majority of the research has focused on the occurrence of EC in alcoholic beverages Due to health risks, some countries have imposed legal limits on carbamate content in alcoholic beverages.

## 1. Introduction

Toxic compounds, harmful to the human body have been identified in fermented foods and beverages over time. Some of them occur naturally as by-products of the fermentation process, such as ethyl carbamate.

Human exposure to these compounds is mainly related to lifestyle and consumption of foods and beverages with high concentration of EC. These high concentrations are the result of improper manufacturing processes [1].

Ethyl carbamate, also known as urethane, is an organic compound, the ethyl ester of carbamic acid, with the chemical formula: CH_3_CH_2_OC(O)NH_2_ that is formed during fermentation, distillation, storage and maturation of alcoholic beverages [2,3,4]. Thus, it has been identified in all products where alcohol is produced through fermentation, such as wine, beer, and distillates, as well as in various fermented foods like yogurt, vinegar, soy sauce, bread, etc.

Ethyl carbamate has been considered carcinogenic and genotoxic since 1974, when the International Agency for Research on Cancer (IARC) and the World Health Organization (WHO), classified it as a Group 2B carcinogen [5]. Moreover, in 2007, was classified as 2A group, meaning “probably carcinogenic to human body” [6,7].

A 1980 study on rats [8], showed that feeding ethyl and vinyl carbamates led to the development of carcinomas in the liver, auditory canal, and ear lobe. Numerous other studies have shown that various tumors, both benign and malignant, occurred in other mammalian species tested with ethyl carbamate [9,10,11]. Other researchers have also demonstrated in their studies, a direct link between liver cancer in humans and EC in food and beverages. Since the mid-20th century studies on the carcinogenic potential of urethane have demonstrated the carcinogenic properties of this compound in developing breast carcinoma, lung adenoma, and blood cysts in the liver [12,13].

Unfortunately, this organic compound is not subject to regulations in all countries. According to the IARC, the average calculated intake of ethyl carbamate from staple foods such as bread and yogurt is about 15 ng/kg bw/day. This value does not include the intake of ethyl carbamate from alcoholic beverages (wine, beer, distillates, and liqueurs). Considering that wines contain about 10–30 μg/L, this is a considerable intake, with some wines far exceeding this limit. Distillates have been found to contain anywhere from a few tens to a few hundred micrograms per liter; some distillates contain around 10 mg/L [14,15].

Moreover, JECFA, an international scientific expert committee, under the Food and Agriculture Organization of the United Nations (FAO) and the WHO, highlighted that the presence of ethyl alcohol in alcoholic beverages increases toxicity and potential carcinogenic effects of ethyl carbamate [16]. Therefore, it is necessary for all countries to regulate ethyl carbamate in wines and distilled alcoholic beverages, as most studies show that human exposure to ethyl carbamate comes primarily from consuming these products.

The risks assessment of the EC exposure was carried out by many government agencies, such as the European Food Safety Authority (EFSA), leading to the establishment of EC content limits in various alcoholic beverages [17].

They took into account the numerous factors that can lead to variations in EC content in foods and beverages, such as: raw materials, microorganisms present, and process parameters [7].

Seven countries, along with a few others in the EU, have imposed a maximum limits on alcoholic and distilled beverages. Most of these countries have set maximum permissible limits on the ethyl carbamate content in fruit brandy. This category includes stone fruit spirits, a distillate for which high and very high ethyl carbamate values have been identified by several researchers in many countries.

Several countries, including Canada, the United States, Brasil, South Korea, and some European countries (such as: Germany, France, and the Czech Republic), have legislation in place regarding the ethyl carbamate content of wines and distillates [18]. These limits are set the following limits for Fruit brandy: 400 μg/kg in Canada and the Czech Republic, 800 μg/kg in Germany and 1000 μg/kg in France. When it comes to distilled spirits (tequila, whiskey, vodka) the allowed limit for EC is 125 μg/kg in the USA, 150 μg/kg in Canada, Czech Republic, and France and 210 μg/kg in Brazil. As far as fortified wines are concerned, the limit allowed for EC varies between 60 and 100 μg/kg (USA and Canada, respectively).

The International Organization of Vine and Wine (OIV) has adopted modern, high-precision methods for detecting ethyl carbamate in wine, but has not set the maximum allowed limits for EC. They only recommend a maximum content of 30 μg/L for the limit of detection of ethyl carbamate in wine [19,20,21].

## 2. Identification in Fermented Beverages and Foods

The ethyl carbamate content of fermented food products varies, for example, low ethyl carbamate values (up to 12 μg/kg) have been identified in bread, which would not normally raise a problem (Table 1) under normal consumption. However, some countries have high bread consumption rates. The annual bread consumption per capita in Romania is over 85 kg [22].

In addition, various additives are usually used as antioxidants, bleaches, and preservatives. This is also the case with potassium bromide, benzoyl peroxide, and especially azodicarbonamide. Azodicarbonamide is an additive not included in the list of additives approved for bread production in the EU, yet it has been found in many finished products, primarily from bakeries and pastry shops owned by investors in countries where these additives are approved. Azodicarbonamide is used as a maturing agent in bakery products to improve their external characteristics. Although it is a toxic, carcinogenic additive, studies have not been able to establish whether azodicarbonamide is the basis for the formation of ethyl carbamate [23].

**Table 1 foods-14-03292-t001:** Ethyl carbamate content of some food products, µg/kg.

No.crt.	Food Products	Value of Ethyl Carbamate µg/kg	Country
1	Bread	1.4–4.8	Canada
		ND–12	Hong Kong, USA [24]
2	Soy sauce	ND–130	China, Japan, USA, South
		8–108	Korea [25,26]
3	Vinegar	ND–51	China, South Korea, the USA, and Eu members
4	Sufu	12–124	China
5	Red sufu	87–344	China

Source: Ryu D. and colab., 2015 [17].

Similar to other fermented foods, higher concentrations of ethyl carbamate have been identified in fermented sauces, especially soy sauce, with maximum concentrations exceeding 130 μg/kg [17,27,28,29]. Maximum values of 124 and 344 μg of ethyl carbamate were also identified in sofu and red sofu. Sofu is a protein-rich, nutritious, and easily digestible product originally from China. It is made from fermented tofu that is aged in brine. Sofu fermentation uses molds from the genera *Mucor*, *Actinomucor*, and *Rhizopus*, as well as bacteria, particularly *Bacillus* spp. [28].

The product is rich in free amino acids due to the hydrolysis of soy proteins and maturation in the presence of ethyl alcohol, which is formed by alcoholic fermentation. This process leads to a high content of volatile aromatic compounds, as well as ethyl carbamate and biogenic amines, such as putrescine and cadaverine. The ethyl carbamate content of foods obtained through alcoholic fermentation, such as dairy products and yogurt, is insignificant and does not pose a problem [30].

## 3. Alcoholic Beverages and Ethyl Carbamate

Ethyl carbamate has been identified in many cases in high and very high concentrations in alcoholic beverages, especially distillates and distillates from stone fruits. Table 2 shows the identified ethyl carbamate content in various distilled alcoholic beverages. Most of the EC values were taken from literature published after the 1990s because, prior to that time, the levels of these chemical compounds could not be accurately determined. Since the 2000s, developments in analytical techniques and methods have facilitated the identification of ECs in very low concentrations, especially in foods and beverages obtained by a fermentative process, such as wine, beer, fermented sauces, and bread [9,31,32,33,34].

Significant levels of ethyl carbamate have been recorded in stone fruit brandy, with concentrations reaching up to 22,000 μg/kg. Lachenmeier DW [49] uses a rapid method involving Fourier-transform infrared (FTIR) spectroscopy combined with partial least squares (PLS) regression to determine the ethyl carbamate content in stone fruit spirits.

The 2007 recommendation by the European Food Safety Authority (EFSA) regarding the reduction of ethyl carbamate and hydrocyanic acid in food and beverages was a crucial step in addressing potential health risks posed by these compounds, particularly in alcoholic beverages made from stone fruits [20]. Ethyl carbamate (also known as urethane) is a carcinogenic compound that can form during the fermentation and distillation of certain foods and beverages, with stone fruit-based spirits being particularly affected due to their high levels of hydrocyanic acid, which is a precursor to ethyl carbamate formation [50,51].

According to the National Institute of Statistics on Beverage Consumption for 2022, the consumption of stone fruit spirits, such as țuica and palinka, is increasing in Romania [52]. Țuica, a traditional Romanian distilled product, is an alcoholic beverage obtained by fermenting and distilling plums; its alcohol concentration is at least 24% by volume. Stone fruit brandy is similar to this drink, but it is made by fermenting and distilling other fruits, such as apricots and stone fruit seeds. Regulations regarding the chemical composition of these distillates mainly concern hydrocyanic acid, imposing a maximum content of 7 g per hectoliter of 100% alcohol.

According to the National Institute of Statistics, average per capita consumption of distilled alcoholic beverages (40% alcohol by volume) increased by 0.5 L in 2022 compared to 2021. Figure 1 shows the evolution of alcoholic beverage consumption in Romania per capita (in liters of 100% alcohol). As can be seen from the graph, beer consumption decreased slightly (1 in Figure 1), wine consumption did not change (2 in Figure 1), and the consumption of distilled alcoholic beverages increased (3 in Figure 1). A significant percentage of this last category is represented by stone fruit spirits, such as țuica and palinka.

Regarding stone fruit brandy, the studies conducted by Ryu D. et al. in 2015 stand out [17]. These studies examined 34 food and beverage products in Korea and the researchers identified an ethyl carbamate content of 151.06 μg/kg in Maesilju, a traditional Korean liqueur made from the green fruits of the Japanese apricot tree (*Prunus mume*), sugar, and rice distillate (named Soju), with an alcohol concentration of about 20%. While the concentration of ethyl carbamate is high, other authors have identified similar concentrations in this Korean alcoholic beverage [53,54]. For example, ethyl carbamate concentrations exceeding 300 μg/kg were identified in Maesilju [55].

Other studies [53,56], on Maesilju have identified a positive correlation between ethyl carbamate content and hydrocyanic acid content, which is a precursor to ethyl carbamate. This correlation is stronger than the correlation with alcohol content. The authors demonstrate that increasing the temperature results in greater formation of ethyl carbamate precursors, and, consequently, a higher ethyl carbamate content in the products [53]. According to results reported in 2013, there is a significant positive correlation between the Maesilju content and alcohol concentration in ethyl carbamate, as well as the maturation duration [38].

Studies conducted on different samples of umeshu have shown the importance of this alcoholic beverage’s storage parameters as well as its cyanide content [57,58]. Umeshu, a very popular Japanese alcoholic beverage, is made by macerating green, unripe ume plums in a distillate of shochu and sugar. Shochu is a distillate made from rice, barley, sweet potatoes, or buckwheat. Sometimes other raw materials, such as carrots, chestnuts, or sesame seeds, are also used. Research has shown increased ethyl carbamate concentrations in samples exposed to light and room temperature compared to those kept in the dark and refrigerated.

In reported research from 2011, an ethyl carbamate content of 33.8 μg/kg was identified in cereal distillates from Hebei Province, China. This content was analyzed using the GC/MS method [59].

Several studies [19,40,42,44], refer to the ethyl carbamate content of brandy and stone fruit brandy (Figure 2). Brandy is an alcoholic beverage with an alcohol concentration between 37.5 and 86% by volume, obtained by aging wine distillates. Stone fruit brandy is an alcoholic beverage distilled from fruits such as apricots, plums or cherries. The term is used for a wide range of fruits but excludes grapes. Studies have identified higher ethyl carbamate values in stone fruit brandy due to the presence of some distillates from stone fruits in this group [18].

Additional research data were reported in 2011 [43] regarding the identification of various ethyl carbamate concentrations in brandy produced in South Africa, with values ranging from 4.4 to 95 μg/L (Table 2). The authors found a positive correlation between the maturation time of the distillate and its ethyl carbamate content. An important role in increasing the ethyl carbamate concentration was also played by the distillates’ temperature and storage duration [43].

Research has also been carried out on other types of distilled beverages, such as whiskey. The maximum ethyl carbamate content identified was in samples of US whiskey (1719 μg/L) and EU whiskey (1000 μg/L) [17,60]. Figure 3 shows the ethyl carbamate values identified in other distilled beverages specific to different countries.

In 2009, a series of studies 2009 on distilled beverages from Mexico, including tequila, mezcal, sotol, bacanora, and cuxa from Guatemala, revealed values of 60 μg/L for cuxa and 390 μg/L for tequila in these distillates [51].

Popular distillates consumed in Brazil include cachaça, obtained from sugarcane and tiquira from cassava root. Studies on popular Brazilian distilled beverages have identified ethyl carbamate content values ranging from 12 to 910 μg/L in infant cane spirits [46,47,61,62]. Cassava root contains hydrocyanic acid, a precursor to ethyl carbamate, which increases in distillates the ethyl carbamate concentrations. These concentrations often exceed the legal limit and need to be subjected to rigorous testing [63].

Figure 4 shows the maximum ethyl carbamate content of various distillates from EU countries found on US markets. The data were submitted by competent organizations in the US following analyses carried out using methods recognized in the US [41]. As expected, the values presented show that fruit brandy, probably mainly stone fruit brandy, exceeds the regulated limit for these drinks.

## 4. Wines Content of Ethyl Carbamate

Table 3 shows the minimum and maximum ethyl carbamate content values identified in different wines from numerous studies carried out in various geographical areas. Data on ethyl carbamate content are presented for normal wine, without other additives, such as etlylic alcohol addition and fortified wines, as well as rice wine and sake.

Studies on the ethyl carbamate content of wines have identified different values depending on their origin (Figure 5). The values are not very high, but concern remains regarding the consumption of this product. Some wine samples analyzed in other studies by different authors from the USA [44] and the EU [20], are found to have a significantly higher ethyl carbamate content than the average.

After analyzing 18 samples of Spanish red wines with a controlled designation of origin, in order to establish correlations between ethyl carbamate and other wine compounds, identified significant correlations between ethyl carbamate content and volatile acidity and ethyl lactate content in wines [64].

Fortified wines are special wines produced by specific methods, usually oxidative, involving the addition of a distillate from grapes during fermentation or later [69,70]. The most popular fortified wines in the world are as follows: Sherry, an oxidatively fortified wine with an alcohol concentration of 15–18%, produced in Spain’s Jerez region from Palomino, Muscat, and Pedro Ximénez grapes; port wine, produced in Portugal’s Douro Valley region; Madeira, a special wine originating in Portugal’s Madeira Islands whose aging period can be several decades; and Marsala, a wine from Sicily, Italy, with an alcohol concentration between 15 and 20%.

Compared to traditional wines, fortified wines such as Porto and Madeira (Figure 6) showed higher concentrations of ethyl carbamate. The maximum values identified were up to 404 μg/L. Wines from the USA stand out again, as do some wines from EU countries [20].

Additionally [43], a direct proportional relationship was found in the study between ethyl carbamate content and alcohol concentration (vol% alcohol) in fortified South African wines as a function of aging time (Figure 7).

## 5. Ethyl Carbamate Formation Mechanism

The main formation routes of ethyl carbamate are as follows:

First, it can form from cyanogenic glycosides, such as amygdalin, which is found in fruit stones (e.g., plums and apricots). These glycosides undergo thermal cleavage and enzymatic degradation to produce hydrocyanic acid. The hydrocyanic acid is then oxidized to produce cyanate, which reacts with ethanol in a reaction catalyzed by copper ions to produce ethyl carbamate [63,71,72];

The second pathway for the formation of ethyl carbamate involves urea accumulated in yeasts. Urea is formed by the degradation of arginine through the action of arginase during fermentation [73]. Urea is further degraded by ureoamidolyase into ammonium and carbon dioxide. Unfortunately, urea is not a preferred nitrogen source for yeasts. When other nitrogen-containing compounds, such as asparagine and glutamine, are present, they are preferred as nitrogen sources, which inactivates the ureoamidolyase enzyme. In this way, urea accumulates in the yeast cell. The cell then releases the urea into the environment, where it reacts with ethanol to form ethyl carbamate [74].

In addition to hydrocyanic acid (cyanide) and urea as a precursor to ethyl carbamate (EC) formation, several other compounds have been identified as potential precursors, making the issue more complex and broad. These precursors can lead to EC formation through various biochemical pathways during fermentation and distillation processes in alcoholic beverages. Here is a brief overview of the key precursors mentioned: carbamylphosphate, citrulline, and diethyl pyrocarbonate [1].

Lactic acid bacteria and yeasts mainly accumulate among these precursors (for example urea, carbamyl phosphate, and citrulline).

## 6. Formation of Ethyl Carbamate in Wines

Ethyl carbamate forms naturally during the alcoholic fermentation of grape must through a reaction between urea produced by yeast and ethyl alcohol. High levels of ethyl carbamate have also been identified in some wines and have been shown to come from crops that were excessively fertilized with nitrogen-based fertilizers containing urea. Thus, the ethyl carbamate content of wines can be directly related to both the grape variety and the agronomic practices applied to the vine, especially the use of nitrogen fertilizers and pesticides.

The explanation lies in the fact that, among the ethyl carbamate precursors, there is also arginine. Arginine is an important amino acid and an intermediate metabolite in the urea and nitric oxide cycles. Many years ago [75], it was demonstrated that vines store a large portion of their nitrogen reserves in the form of low-molecular-weight, soluble compounds, such as amino acids and amides (especially arginine), in their perennial, woody components (stem and root). Several researchers have shown that between 50 and 70% of the vine’s stored soluble nitrogen is arginine. This amino acid—arginine, provides the vine with necessary nitrogen in the spring when it begins to grow. Excessive fertilization is a common problem in vine plantations and leads to high levels of arginine, an important precursor of ethyl carbamate.

In 2017, a team from Chile and Spain published research results about their study of the accumulation of different amino acids in Cabernet Sauvignon, Pinot Noir, Muscat Gordo, Grenache, Riesling, and Sangiovese grapes, which used high-performance liquid chromatography (HPLC) analysis [76]. They observed differences in free amino acid content between varieties, but proline and arginine were the major amino acids in all varieties. Regarding the distribution of arginine in the analyzed grapes (Cabernet Sauvignon and Riesling), higher values were found in the grape berry epicarp compared to the mesocarp. This explains why red wines have a higher ethyl carbamate content than white wines [76].

Research conducted in Australian vineyards: shows that arginine levels in vines are better indicators of nitrogen content than total nitrogen levels [77]. In the same context, the authors demonstrate that yeast and lactic acid bacteria growth and multiplication processes, as well as alcoholic and malolactic fermentation kinetics during winemaking, are related to must nitrogen content. An important fermentation parameter is amino acid concentration. One easy way to increase the nitrogen compound content in the must is to fertilize vine plants with urea, especially through foliar application, a technique that leads to rapid assimilation by the plant [75,77].

Like other researchers, it was demonstrated in other studies [78,79], that excessive nitrogen fertilization, especially foliar application, affects grape composition and leads to higher concentrations of ethyl carbamate in wines. Apart from vine variety and the agrotechnics applied to the plant, several other factors affecting ethyl carbamate content in wine have been identified. The optimization of process parameters, such as temperature, yeast species and strain, etc., is an identified factor, albeit of lesser importance. Conversely, numerous studies show that significant increases in ethyl carbamate content occur during wine maturation or storage due to reactions between ethanol and different wine compounds, depending on temperature, light, etc. [80].

Studies conducted, on rice wine Hakka Huangjiu a popular beverage in the Jiangnan area, show changes in ethyl carbamate and its precursors, including citrulline, arginine, ornithine, and urea, with increasing ripening temperature [81]. The results indicate an increase in both ethyl carbamate and citrulline concentrations with increasing ripening temperature, though arginine and ornithine concentrations slightly decreased [81].

One of the EC precursors in wines is diethyl pyrocarbonate [5], a chemical compound often used as an antiseptic additive to inhibit the growth of pathogenic microorganisms, similar to sulfur dioxide. It reacts with ammonia to form EC [82]. Fortunately, most countries have banned this substance from winemaking due to its toxicity and side effects [83,84].

## 7. Discussion

### 7.1. Bibliometric Analysis

To assess the state of knowledge on the topic of ethyl carbamate (EC) in fermented foods and beverages, a bibliometric analysis was conducted using data from the Scopus database, a renowned scientific research platform from Elsevier. A search using the keyword combination “ethyl” AND “carbamate” resulted in the identification of 2058 documents published between 1945 and 2025. Among these, a peak of 153 articles was published in 2021. In contrast, a search on Google Scholar yielded over 17,200 results (1989–2025) related to the same keywords.

However, a more focused search using the keyword combination “ethyl” and “carbamate” and “fermented” and “beverages” yielded only 27 scientific documents. The year of publication of the first article among the 27 documents is 1989 [30], and the last included was published in April 2025 [85].

Dennis et al. [30], in 1989 publication, described an analytical procedure to detect ethyl carbamate traces in fermented foods and alcoholic beverages. The 27 scientific papers mentioned on the Scopus platform following keyword searches were published by authors from eight countries, such as China and North Korea, who were in the top two spots with 14 and six publications, respectively; the Netherlands and the United Kingdom followed with two publications each; and Brazil, Denmark, Germany, and Japan followed with one publication each. This bibliometric analysis indicates a growing interest in this topic within the scientific community.

### 7.2. Minimizing EC in Fermented Products

Several studies have explored methods to reduce EC concentrations in fermented products. One promising technique involves the incorporation of molecularly imprinted polymers (MIPs) into filtration systems and solid-phase extraction processes [85]. However, the use of MIPs in alcoholic beverage production needs further investigation due to potential environmental risks posed by organic solvents and other chemical substances if the production process is not managed efficiently. The research team that published the most recent study in 2025 recommended further research on multiple matrices to ensure consumer safety [85].

Another recent report (2025) [86] provides an overview of the advantages and disadvantages of the most commonly used polymerization approaches for MIP synthesis. The report also discusses the analytical techniques employed for MIP characterization and strategies to mitigate EC in fermented alcoholic beverages, with some studies reporting removal efficiencies of up to 84%.

Fermented alcoholic beverages, especially wine, are the main sources of ethyl carbamate. Wine is the second most widely consumed fermented beverage worldwide, following beer [87]. Research focusing on reducing EC levels in wine shows that modifying the fermentation microorganisms is crucial to achieve this goal. Therefore a significant reduction in EC can be achieved by modifying genes that are involved in the metabolism of arginine and urea. Moreover, when optimizing raw material processing and fermentation conditions EC content is reduced. Enzymatic methods, such as the use of urease or urethanase to degrade EC precursors, reduce EC formation even further during wine maturation and aging. Research conducted by a Chinese team in 2023 [87] demonstrated that genetic engineering of wine yeast strains can significantly reduce EC levels by modifying genes involved in arginine and urea metabolism.

Similar strategies have been explored for controlling EC in other fermented foods. These include the use of genetically modified microorganisms in the fermentation process, enzymatic degradation of EC, and physical adsorption using activated carbon. Controlling raw material composition, such as limiting excessive nitrogen in vineyards, is also critical in reducing EC formation [88].

### 7.3. Detection Methods for EC in Fermented Foods and Beverages

The detection of ethyl carbamate is mandatory for characterizing fermented foods and establishing acceptable EC limits. Recent studies have led to the development of novel detection methods, including the following:

Opti sensors: Han et al. (2022) [89,90] developed an optisensor based on core-shell nanostructures of carbon dots derived from watermelon rind biomass, integrated with MIPs via bulk and sol-gel methods. This sensor showed high optical properties and specific adsorption of EC, offering good linearity (1–120 μg/L), low detection limits (0.57–0.94 μg/L), and fast detection times [89,90,91].

Photoelectrochemical Biosensors: Another study [92] developed a non-contact biosensor using hierarchical MXene/Bi_2_S_3_ nanolayers as the sensing material. The biosensor detects ammonia produced by an enzyme-coupled reaction, which modifies the photocurrent of the material. Signal amplification through hyperbranched DNA assembly and enzyme encapsulation leads to a significant photocurrent change for detecting high EC concentrations in fermented foods and beverages.

### 7.4. EC in Distilled and Fermented Beverages

Baijiu, a traditional Chinese distilled alcoholic beverage, is characterized by its high alcohol content (around 55% by volume) and diverse flavour profile. Research has shown that the presence of EC in Baijiu decreases its ester content by activating hydrolysis, a process accelerated by light and heat exposure during storage [93]. Studies on other Chinese distilled beverages have demonstrated that EC is formed during fermentation, distillation, and storage, and there are similar strategies for reducing EC content as those used in fermented beverages [94].

### 7.5. Health Implications and Carcinogenic Potential

Research on rats treated with Musalais, a wine fermented from Hetianhong grape must, has shown that high EC concentrations are associated with carcinogenic potential. The study identified differentially expressed genes (DEGs) in rats treated with Musalais, indicating that EC influences cytochrome P450 metabolism, chemical carcinogenesis, and xenobiotic metabolism via cytochrome P450 and Wnt signaling pathways, leading to cancer development [95].

Further studies have examined the formation and metabolism of EC in fermented foods and its genotoxic potential. Strategies to reduce EC content involve physical, chemical, enzymatic, and genetic engineering methods applied to fermentation microorganisms [96].

### 7.6. Quantitative Detection Methods

Several quantitative detection methods for EC in fermented foods have been developed, including the following:

Chromatographic Methods (GC and LC): These methods, often coupled with mass spectrometry (MS) or fluorescence detection, are considered the highest standard for EC quantification due to their sensitivity and accuracy [97]. Those methods [17,98] were used to determine the concentration of EC in fermented foods and beverages such as apple juice, milk, corn oil, soybeans, rice porridge, peanut butter, fish, beef, and sea mustard. These products were also analyzed using gas chromatography coupled with mass spectrometry (GC-MS) [98,99].

Alternative Methods: Methods such as enzyme-linked immunosorbent assays (ELISA), enzymatic tests, and nanosensors have been developed to provide high specificity or rapid screening for EC [97].

An example of a specific application is the analysis of cachaça, a traditional Brazilian alcoholic beverage, using high-performance liquid chromatography with fluorescent detection (HPLC-FLD). This method has been validated for its sensitivity, specificity, good linearity, and low detection limits, making it suitable for detecting trace levels of EC in fermented alcoholic beverages [100].

### 7.7. Regional Exposure to EC

Studies on EC exposure in specific populations have shown low overall risk, though continuous monitoring is essential. For example, a study on the Korean population’s exposure to EC in fermented foods and beverages found that daily EC intake for frequent consumers was 12.37 ng/kg body weight [56]. Despite the relatively low levels of EC in traditional Korean fermented beverages, such as takju and yakju, the concentrations of EC in some fermented products, including Korean plum wine and sake, were found to be high, requiring careful monitoring [101].

Romania has many traditional alcoholic and non-alcoholic fermented beverages, such as braga, borș, and socata, which have many nutritional properties, but only recently have Romanian researchers started studying them to identify and quantify any undesirable, potentially toxic compounds they may contain [102].

## 8. Conclusions

Ethyl carbamate remains a significant concern in fermented foods and beverages due to its potential carcinogenic effects.

In Eastern countries, including Romania, stone fruit brandy, brandy, and spirits such as “palinka” are distillates that are consumed in large quantities, and their consumption has increased in recent years. Clear regulations regarding the ethyl carbamate content of these distillates are thus required.

Lower concentrations of ethyl carbamate have been identified in traditional wines. However, the agronomic practices used in the vineyard must be considered, as excessive nitrogen fertilization leads to higher concentrations of ethyl carbamate in wines. Compared to traditional wines, fortified wines had a higher ethyl carbamate content, which was positively correlated with the wines’ alcohol content, as well as with their aging time and storage conditions.

While various strategies, such as genetic modification of microorganisms, enzymatic degradation and the use of molecularly imprinted polymers, show promise for reducing EC concentrations, more research is needed to ensure safety, especially in high-risk beverages. Moreover, advances in detection methods, mainly novel biosensors and chromatographic techniques, are vital for monitoring EC levels. Continuous global efforts to better understand and reduce the risks associated with EC in fermented products are mandatory for ensuring consumer safety.

## Figures and Tables

**Figure 1 foods-14-03292-f001:**
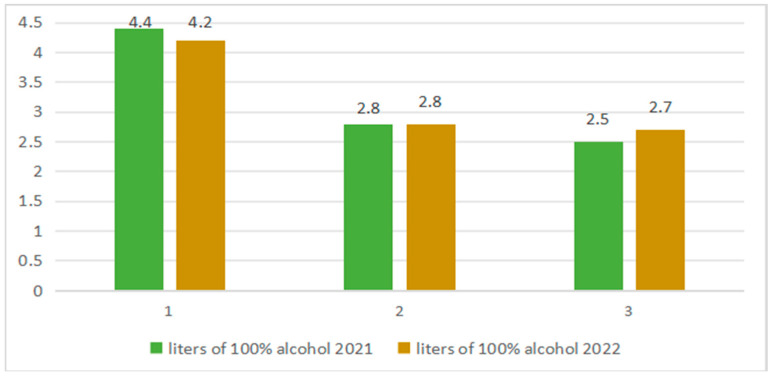
Evolution of average alcohol consumption per capita, in 2021–2022 (liters of 100% alcohol)**.** Source: National Institute of Statistics Romania, 2022 [52].

**Figure 2 foods-14-03292-f002:**
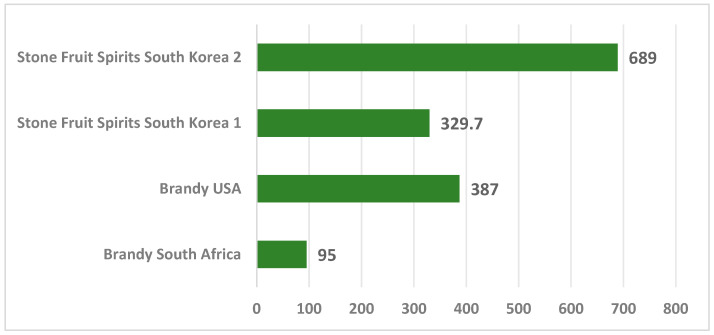
Maximum ethyl carbamate content identified in brandy and stone fruit brandy, in various countries (µg/L ethyl carbamate); Source: [18,19,40,42,44].

**Figure 3 foods-14-03292-f003:**
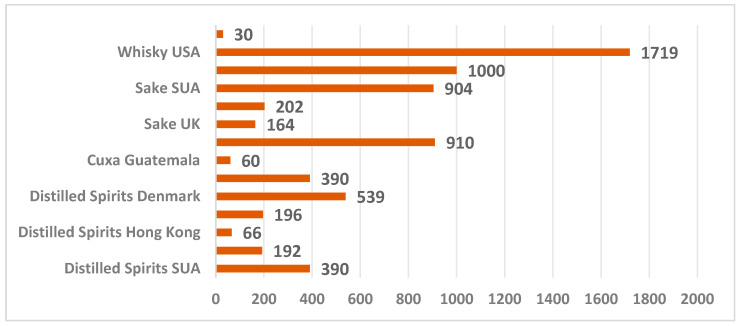
Maximum ethyl carbamate content identified in various distilled beverages and sake, in various countries (μg/L).

**Figure 4 foods-14-03292-f004:**
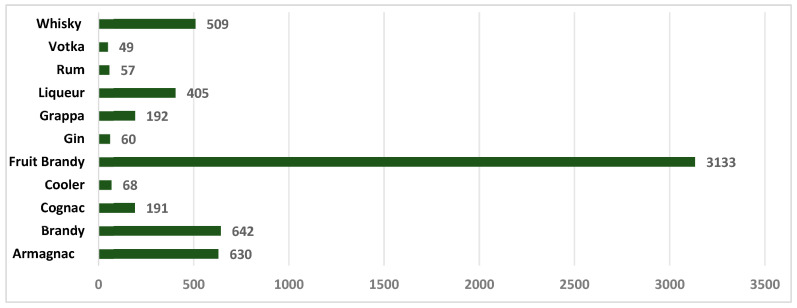
Maximum ethyl carbamate content (µg/kg) identified in various distilled beverages originating from the EU, found on the US market.

**Figure 5 foods-14-03292-f005:**
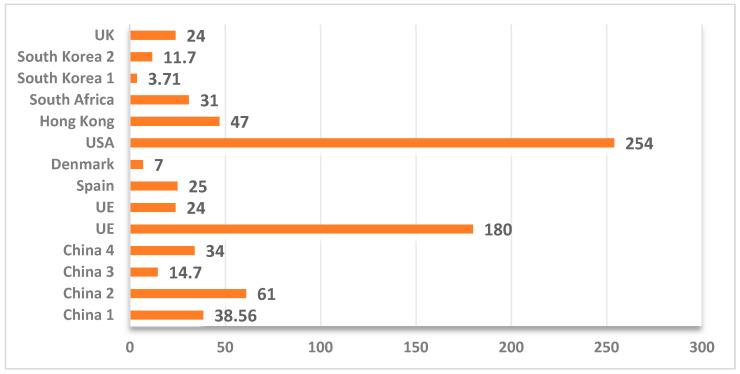
Maximum ethyl carbamate content identified in different wines without additives, in various countries (μg/L).

**Figure 6 foods-14-03292-f006:**
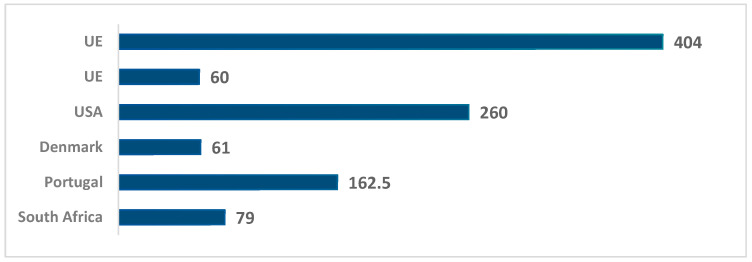
Maximum ethyl carbamate content identified in various fortified wines, in various countries (μg/L).

**Figure 7 foods-14-03292-f007:**
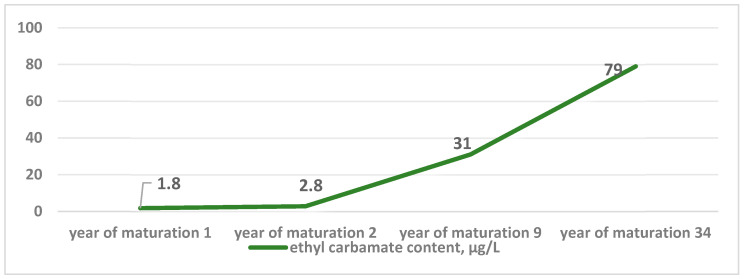
Correlation between ethyl carbamate content of wines and their aging period (μg/L).

**Table 2 foods-14-03292-t002:** Values of ethyl carbamate identified in various alcoholic beverages, from various countries (µg/L ethyl carbamate).

Alcoholic Beverages	Value of Ethyl Carbamate µg/ L	Country	References
Distilled spirits	5–5000	Denmark	[35]
	192	China	[36]
	20–66	Hong Kong	[37]
	196	South Korea	[17,38,39]
	390	USA	[19,40]
Stone fruit spirits	22,000	UE	[16,17]
	18,000	Germany	[28]
Whisky	ND-1000	UE	[17,41]
	1719	USA	[17]
	30	Korea	[42]
Maesil	67.9	Korea	[38]
	151.06	Korea	[17]
Brandy	4.4–95	South Africa	[43]
	387	USA	[17,44]
Fruit brandy	5100	USA	[19,40]
	329.7	Korea	[19,42]
	689	Korea	[42]
Agave spirits (tequila)	390	Mexico	[19,45]
Cuxa	60	Guatemala	[19,45]
Sugar cane spirits (cachaҫa)	12–910	Brasil	[46,47,48]

**Table 3 foods-14-03292-t003:** Concentration values of ethyl carbamate (μg/kg) identified in different wines and countries.

Alcoholic Beverage	Values	Country	References
Wines without additives	ND–25	Spain	[64]
	ND–24	UE	[17,19]
	ND–180	China	[41]
	1.8–31	South Africa	[43]
	254	USA	[17,44]
	5–7	Denmark	[24]
	11–24	UK	[65]
	6–47	Hong Kong	[17,37]
	3.71–11.7	South Korea	[17,42]
	1.16–38.56	China	[17,36,66]
	61	China	[19]
Fortified wines	1.8–31	South Africa (age 1–9)	[43]
	2.8–79	South Africa (age 2–34)	[43]
	54.1–162.5	Portugal (Madeira)	[67]
	7–61	Denmark	[35]
	14–60	UE	[17,65]
	ND–404	UE	[20]
	ND–260	USA	[19]
Rice wine	8–515	China	[36]
	ND–580	China	[19]
	242.2	China (yellow rice wine)	[68]
	14.11	South Korea	[17,38,39]
Sake	81–164	UE members	[17,20]
	ND–202	Japan	[20]
	904	USA	[17,44]

## Data Availability

No new data were created or analyzed in this study. Data sharing is not applicable to this article.

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
