# Peer review of "Studies on the Ethyl Carbamate Content of Fermented Beverages and Foods: A Review"

_foods, 2025, doi:10.3390/foods14193292_

Round 1
Reviewer 1 Report
Comments and Suggestions for Authors
- Current early detection technologies present several limitations that require further refinement. To enhance the analytical framework, it is recommended to incorporate recent literature, cited from 1990 onwards.
- The clarity of the results could be significantly improved through advanced data visualizations, such as the regulatory comparison illustrated in Figure 1 and the content distributions detailed in Figures 3-8. Furthermore, the reproducibility of the literature review would be strengthened by explicitly stating the screening criteria, including the specific databases searched and the keywords used. While the existing dataset draws from samples in Europe, America, and Asia, it lacks sufficient representation from Latin America and Africa, limiting the generalizability of the findings.
- The analysis would also benefit from the inclusion of additional control strategies, for instance, yeast strain screening and precursor degradation technologies.
- Regarding causal inference, caution is warranted; the relationship between ethyl carbamate (EC) and cancer should be emphasized as a correlation, in alignment with the IARC statement, rather than an absolute causal link
- To improve terminological precision, "cattamates" should be corrected to "carbamates," and the term "stone fruit spirits" must be used consistently throughout.
- It is also advised to add a dedicated 'Discussion' section to explicitly address the study's limitations and propose clear directions for future research
To improve terminological precision, "cattamates" should be corrected to "carbamates," and the term "stone fruit spirits" must be used consistently throughout.
Author Response
Thank you for the review of our manuscript. We have revised the manuscript in response to your suggestions.
Our response is provided below .
Authors Response:
1
Regarding the request to add applications: We have added a review of applications as Chapter 7. Please see this addition on pages 13-16 of the revised manuscript.
2
The results are now significantly clearer thanks to the comments accompanying the illustrations in Figures 1 and 3-8.
In order to assess the state of knowledge on the subject of this review, a bibliometric analysis was conducted using data from the Scopus database, which belongs to the renowned international scientific research platform Elsevier.
As you noted, most of the data reported in our review was published after studies on fermented beverages and foods in Asia were conducted.
However, we also highlighted research on various fermented products from countries such as Brazil, Mexico, Guatemala, Chile, and Africa in our references.These references were added by us in Table 2 and comments betwen line 229-237, also in Figure 6.
3
Taking your comments into account, a new series of references has been added to Chapter 7. These references [86, 87, 88], are related to the inclusion of additional control strategies and the modification of genetically modified fermentation microorganisms. EC reduction is achieved by modifying the genes involved in the metabolism of precursors.
4
Through our review, we found that results published in Cancer Research [reference 8 and 13] (references 8 and 13), Seminars in Cancer Biology [reference 62], and BMC Cancer [reference 63] provide evidence of a direct link between ethyl carbamate (EC) and cancer.
5
Thank you for your comments. As a result, we have revised the article by adding the specialized terms you mentioned.
6
In accordance with your recommendation, we have developed a new chapter, Chapter 7: 'Discussion', in which we highlight innovative elements in the field and comment on 17 recent references. For example, we mentioned research on minimizing the level of ethyl carbamate through the use of genetically modified microorganisms, pretreatment of raw materials, and/or enzymatic methods.
7
Thank you for your comments. As a result, we have revised the article by adding the specialized terms you mentioned. We have also worked on a more accurate translation of the article in order to improve its readability.
Reviewer 2 Report
Comments and Suggestions for Authors
The authors review studies on ethyl carbamate occurrence in foods and beverages.
The following revisions might be considered:
- What is the novelty of the review compared to the many previous reviews that are cited?
- A method section should provide the search and review strategy. I think many studies on EC occurrence have been missed and the coverage is unclear. Did you include newer literature not reviewed before?
- Throughout: English and Grammar must be improved by a native speaker
- Please state line numbers for peer review (check journal template)
- Introduction, end of first page: Formaldehyde is not in the same class as ethyl carbamate (formaldehyde is IARC group 1 and not 2A).
- Throughout: the intext references must be adjusted to journal style. Drop all first names and only show the first author with et al. and reference number, e.g. Gowd et al. [1]. Do not provide year for intext references.
- Throughout: It must read “et al.” and not “et all.”
- Page 2, 2nd line: I do not believe that food cocentrations lead to serious neurological conditions, recheck
- Page 2; avoid indirect citations such as “also by Zhao”
- The FAO is independent and not of the WHO. What you mean here is the JECFA, please correctly refer to JECFA
- Please re-check the claim about ethyl alcohol potentiating the effect of ethyl carbamate by factor 4
- The EFSA opinion might be included already in the introduction
- Figure 1 and corresponding text. The figure is barely readable and the regulatory information is mainly outdated. E.g. Germany never had a real limit and this was superseded by the EU recommendation (which is true for all EU countries)
- Page 4: I would delete an unqualified information such as “can have toxic effects on the human body”
- Table 2: check spelling of Mexico
- Page 5: the EFSA does not make regulations. This was an opinion. Afterwards, the EU commission may make an regulations, but in the case of EC this is only a recommendation. Please also cite this document.
- Throughout: in the EU context (e.g. page 5), the term brandy is restricted to wine-based distillates. Check the EU regulation on spirit drinks and their defintions. The products are not called fruit brandies, but fruit spirits or stone-fruit spirits. As this is a paper from Romania, an EU country, I would highly recommend using the nomenclature of the EU spririts regulation.
- Page 6, line 4: brandies and spirits are not necessarily stone fruit spirits. This is a strange sentence
- Figure 2: the decimal in the axis should be point not comma. The in figure legends should be in English.
- Page 6, what is “seed distillate”?
- Page 6 bottom: µ/kg. g is missing?
- Page 7: distinguish brandy and fruit brandy? Is this only wine-based brandy in this case?
- Page 8, delete unqualified claim “dangerous to health”. A quantitative risk assessment would be necessary.
- Table 3: I would only state “EFSA” as author and not “EFSA journal”
- What is wine in stricto sensu
- Figure 9: this is somehow scrambled?
The English is difficult to understand
Author Response
Comments and Suggestions for Authors
The following revisions might be considered:
- What is the novelty of the review compared to the many previous reviews that are cited?
- Authors Response:
This review is based on an extensive compilation of literature providing a comprehensive analysis of ethyl carbamate levels in fermented beverages and foods. While previous studies have addressed ethyl carbamate in food products, some lacked comprehensive analyses summarizing its quantitative presence. Therefore, we consider this work to be of both theoretical and practical significance, as it introduces new approaches to the subject.
- A method section should provide the search and review strategy. I think many studies on EC occurrence have been missed and the coverage is unclear. Did you include newer literature not reviewed before?
- Authors Response:
Through our review, we found that results published in Cancer Research [reference 8 and 13] (references 8 and 13), Seminars in Cancer Biology [reference 62], and BMC Cancer [reference 63] provide evidence of a direct link between ethyl carbamate (EC) and cancer.
- Throughout: English and Grammar must be improved by a native speaker
- Authors Response:
Thank you for your feedback, which has prompted us to completely revise the article.
- Please state line numbers for peer review (check journal template)
- Authors Response:
We have provided for the necessary revision in this revised version, with this aspect highlighted by you.
- Introduction, end of first page: Formaldehyde is not in the same class as ethyl carbamate (formaldehyde is IARC group 1 and not 2A).
- Authors Response:
We have provided for the necessary revision in this revised version, with this aspect highlighted by you.
- Throughout: the intext references must be adjusted to journal style. Drop all first names and only show the first author with et al. and reference number, e.g. Gowd et al. [1]. Do not provide year for intext references.
- Authors Response:
We have provided for the necessary revision in this revised version, with this aspect highlighted by you.
- Throughout: It must read “et al.” and not “et all.”
- Authors Response:
We have provided for the necessary revision in this revised version, with this aspect highlighted by you.
- Page 2, 2nd line: I do not believe that food cocentrations lead to serious neurological conditions, recheck
- Authors Response:
Thank you for your feedback. We have made the changes accordingly. Please see lines 42-50.
- Page 2; avoid indirect citations such as “also by Zhao”
- Authors Response:
We have provided for the necessary revision in this revised version, with this aspect highlighted by you.
- The FAO is independent and not of the WHO. What you mean here is the JECFA, please correctly refer to JECFA
- Authors Response:
Thank you for your feedback. We have made the changes accordingly. Please see lines 58-61.
- Please re-check the claim about ethyl alcohol potentiating the effect of ethyl carbamate by factor 4
- Authors Response:
We have provided for the necessary revision in this revised version, with this aspect highlighted by you. Please see lines 61.
- The EFSA opinion might be included already in the introduction
- Authors Response:
We read and corrected the manuscript based on your suggestions.
- Figure 1 and corresponding text. The figure is barely readable and the regulatory information is mainly outdated. E.g. Germany never had a real limit and this was superseded by the EU recommendation (which is true for all EU countries)
- Authors Response:
Thank you for your feedback. We have made the changes accordingly. Please see lines 75-77.
- Page 4: I would delete an unqualified information such as “can have toxic effects on the human body”
- Authors Response:
We have taken your suggestion into consideration. As a result, we revised it by deleting the information.
- Table 2: check spelling of Mexico
- Authors Response:
We have taken your suggestion into consideration. We reviewed and corrected the information.
- Page 5: the EFSA does not make regulations. This was an opinion. Afterwards, the EU commission may make an regulations, but in the case of EC this is only a recommendation. Please also cite this document.
- Authors Response:
Thank you for your feedback. We have made the changes accordingly. Please see lines 141-149.
- Throughout: in the EU context (e.g. page 5), the term brandy is restricted to wine-based distillates. Check the EU regulation on spirit drinks and their defintions. The products are not called fruit brandies, but fruit spirits or stone-fruit spirits. As this is a paper from Romania, an EU country, I would highly recommend using the nomenclature of the EU spririts regulation.
- Authors Response:
We revised by correcting the information in Table 2, under the term Fruit spirits, in accordance with the articles cited.
- Page 6, line 4: brandies and spirits are not necessarily stone fruit spirits. This is a strange sentence
- Authors Response:
Thank you for your feedback. We have made the changes accordingly. Please see lines 150-154.
- Figure 2: the decimal in the axis should be point not comma. The in figure legends should be in English.
- Authors Response:
We have taken your suggestion into consideration. We reviewed and corrected the information.
- Page 6, what is “seed distillate”?
- Authors Response:
Thank you for your feedback. We have made the changes accordingly. Please see lines 153-154.
- Page 6 bottom: µ/kg. g is missing?
- Authors Response:
We have taken your suggestion into consideration. We reviewed and corrected the information.
- Page 7: distinguish brandy and fruit brandy? Is this only wine-based brandy in this case?
- Authors Response:
Thank you for your feedback. We have made the changes accordingly. Please see lines 197-203.
- Page 8, delete unqualified claim “dangerous to health”. A quantitative risk assessment would be necessary.
- Authors Response:
We have taken your suggestion into consideration. We reviewed and corrected the information.
- Table 3: I would only state “EFSA” as author and not “EFSA journal”
- Authors Response:
We have taken your suggestion into consideration. We reviewed and corrected the information.
- What is wine in stricto sensu
- Authors Response:
Thank you for your feedback. We have made the changes accordingly. Please see lines 244-247.
- Figure 9: this is somehow scrambled?
- Authors Response:
We did everything we could to improve the figures, and we hope it is now clearer.
Comments on the Quality of English Language
The English is difficult to understand
Authors Response:
Thank you for your comments. As a result, we have revised the article by adding the specialized terms you mentioned. We have also worked on a more accurate translation of the article in order to improve its readability. The errors were reviewed and fixed as needed. The reviewer is correct and the manuscript has been checked with English grammar - https://www.deepl.com/en/write.
Reviewer 3 Report
Comments and Suggestions for Authors
Comments on “Studies on The Ethyl Carbamate Content of Fermented Beverages and Foods: A Review”
The authors have compiled an extensive body of literature, providing a comprehensive review of ethyl carbamate levels in fermented beverages and foods. While numerous studies have addressed ethyl carbamate in food products, there remains a gap in comprehensive reviews specifically summarizing its quantitative presence. This work therefore holds practical significance.
Minor Recommendations:
Figure 1, Requires redesign. The x-axis font size is excessively small, rendering it illegible.
Figure 9, The image appears corrupted. Please regenerate it, and consider incorporating chemical structures of key compounds. The current flowchart lacks sufficient detail and depth.
Other Figures. All other figures exhibit suboptimal font clarity. Please enhance resolution and legibility.
Many references are outdated. Please supplement with studies from the past three years to strengthen current relevance.
Author Response
Thank you for the review of our manuscript. We have revised the manuscript in response to your suggestions.
Our response is provided below.
Authors Response:
In accordance with your recommendation, we have developed a new chapter, Chapter 7: 'Discussion', in which we highlight innovative elements in the field and comment on 17 recent references. For example, we mentioned research on minimizing the level of ethyl carbamate through the use of genetically modified microorganisms, pretreatment of raw materials, and/or enzymatic methods.
The results are now significantly clearer thanks to the comments accompanying the illustrations in Figures 1 -9.
As a result, of your comments we have revised the article by adding the specialized terms you mentioned. We have also worked on a more accurate translation of the article in order to improve its readability.
We want to thank you again for all your support.
Submission Date - 17 August 2025
Date of this review-05 September 2025
Reviewer 4 Report
Comments and Suggestions for Authors
This manuscript provides a timely and relevant review of ethyl carbamate in fermented products, addressing both health concerns and regulatory aspects. The topic is of significant interest to the food safety and toxicology communities. The literature coverage is extensive, and the data compilation from various geographic regions is commendable. However, the manuscript requires substantial revisions to enhance its clarity, structure, and scientific rigor.
- Reorganize sections to improve logical flow. For example, Section 5 ("Ethyl carbamate formation mechanism") should precede Sections 3 and 4 to provide context for the reported EC levels.
- Combine redundant paragraphs (e.g., multiple paragraphs on stone fruit spirits).
- Improve figure quality: Axes labels, legends, and data sources must be clearly legible.
- Tables (especially Table 2) have inconsistent formatting (misplaced country/reference entries, e.g., "192 China" and "Ryu D., et al, 2015 [17]" are unaligned). Which country does UE refer to? European Union? Why are there two “UE” in Figure 7?
- Consider consolidating Tables 2 and 3 for better readability.
- Provide more critical insight into the data. For example, discuss why certain beverages (e.g., stone fruit spirits) have exceptionally high EC levels compared to others.
- While EC formation factors are discussed, practical methods to reduce EC (e.g., raw material pretreatment, yeast strain selection) are minimally addressed, limiting the review’s utility for producers.
- Expand the discussion on the health risk assessment of EC, including exposure levels and risk management approaches in different regions. Compare and contrast regulatory frameworks more critically.
- Strengthen the conclusion by summarizing key findings and highlighting future research directions (e.g., need for low-EC fermentation technologies, global harmonization of regulations).
- Most citations are pre-2022; few 2022–2024 studies are included, reducing timeliness for a 2025 submission. Check consistency in reference formatting (e.g., some references lack DOIs or full journal names).
Author Response
Dear Scientific Editor: Reviewer #4:
Thank you for the review of our manuscript. We have revised the manuscript in response to your suggestions. Our response is provided below.
Comments and Suggestions for Authors
- Reorganize sections to improve logical flow. For example, Section 5 ("Ethyl carbamate formation mechanism") should precede Sections 3 and 4 to provide context for the reported EC levels.
Authors Response:
In accordance with your recommendation, we have developed a new chapter, Chapter 7: Discussions. In this chapter, we highlight the innovative elements addressed and comment on them using 17 recent references that report specific EC level values using innovative techniques.
- Combine redundant paragraphs (e.g., multiple paragraphs on stone fruit spirits).
Authors Response:
We have provided for the necessary revision in this revised version, with this aspect highlighted by you.
3. Improve figure quality: Axes labels, legends, and data sources must be clearly legible.
Authors Response:
We did everything we could to improve the figures, and we hope it is now clearer.
4. Tables (especially Table 2) have inconsistent formatting (misplaced country/reference entries, e.g., "192 China" and "Ryu D., et al, 2015 [17]" are unaligned). Which country does UE refer to? European Union? Why are there two “UE” in Figure 7?
Authors Response:
Figure 7 shows two histograms for the EU because EFSA (acording to reference no.20) reports different ethyl carbamate concentration values for fortified wines.
5. Consider consolidating Tables 2 and 3 for better readability.
Authors Response:
In accordance with your we did everything we could to improve the all the Tables, and we hope it is now clearer.
6. Provide more critical insight into the data. For example, discuss why certain beverages (e.g., stone fruit spirits) have exceptionally high EC levels compared to others.
Authors Response:
In accordance with your recommendation, now we revised In the line 145-149 with explain this process
7. While EC formation factors are discussed, practical methods to reduce EC (e.g., raw material pretreatment, yeast strain selection) are minimally addressed, limiting the review’s utility for producers.
Authors Response:
In accordance with your recommendation, we have developed a new chapter, Chapter 7: Discussions, in which we highlight innovative elements in the field and comment on 17 recent references. For example, we mentioned research on minimizing the level of ethyl carbamate through the use of genetically modified microorganisms, pretreatment of raw materials, and/or enzymatic methods.
8. Expand the discussion on the health risk assessment of EC, including exposure levels and risk management approaches in different regions. Compare and contrast regulatory frameworks more critically.
Authors Response:
We have provided for the necessary revision in this revised version, with this aspect highlighted by you.
9. Strengthen the conclusion by summarizing key findings and highlighting future research directions (e.g., need for low-EC fermentation technologies, global harmonization of regulations).
Authors Response:
In accordance with this recommendation, we have created a new chapter, number 7, titled "Discussions." In this chapter, we highlight innovative elements in the field through seventeen recent references that comment on new directions in research studied worldwide.
10. Most citations are pre-2022; few 2022–2024 studies are included, reducing timeliness for a 2025 submission. Check consistency in reference formatting (e.g., some references lack DOIs or full journal names).
Authors Response:
Thank you for your feedback. As a result, we revised the article by adding data from publications of the last three years. We have also completely revised the list of DOIs and journal names in the References section.
Comments on the Quality of English Language
To improve terminological precision, "cattamates" should be corrected to "carbamates," and the term "stone fruit spirits" must be used consistently throughout.
Authors Response:
Thank you for your comments. As a result, we have revised the article by adding the specialized terms you mentioned. We have also worked on a more accurate translation of the article in order to improve its readability.
We want to thank you again for your support.
Submission Date - 17 August 2025
Date of this review-05 September 2025
Round 2
Reviewer 2 Report
Comments and Suggestions for Authors
I still believe that the regulatory information shown in Figure 1 is outdated and I would suggest to delete the FIgure.
Figure 9 contains a panel "double click to edit" which overlays the text, which is very low in resolution
Reviewer 4 Report
Comments and Suggestions for Authors
1. What is the specific value in Figure 8 of "year of maturation 2"?
2. Figure 9 needs to be redrawn. The current version is not clear enough, and there is unnecessary information in the figure.
3. The references numbered 21, 22, 52, 70 and 84 seem to not meet the requirements of the "Foods" journal.
Author Response
Dear Scientific Editor: Reviewer 4:
Thank you for the review of our manuscript. We have revised the manuscript in response to your suggestions. Our response is provided below.
Comments and Suggestions for Authors
Thank you for your comments. As a result, we have revised the article to make it more accurate and readable. We reviewed and fixed the errors as needed. We hope that the manuscript is now better checked for English language errors.
We would greatly appreciate your consideration of the manuscript.
Kind regards.
